# Prevalence of Low Back Pain and Associated Risk Factors among Nurses at King Abdulaziz University Hospital

**DOI:** 10.3390/ijerph18041567

**Published:** 2021-02-07

**Authors:** Aishah Almaghrabi, Fatmah Alsharif

**Affiliations:** 1Faculty of Nursing, King Abdulaziz University, Jeddah 21589, Saudi Arabia; 2Department of Medical Surgical Nursing, Faculty of Nursing, King Abdulaziz University, Jeddah 21589, Saudi Arabia; falsharif@kau.edu.sa

**Keywords:** prevalence, low back pain, risk factors, nurses

## Abstract

Aim: To determine the prevalence of LBP and the associated risk factors among nurses at King Abdulaziz University Hospital (KAUH). Methods: A cross-sectional study design was adopted with a convenience sample of 234 nurses recruited from nine different departments at KAUH in Jeddah, Saudi Arabia. Participants completed the questionnaire, which had two parts: Part I: Socio-demographic data, medical factors, and work-related factors; and Part II: Standardized Nordic Musculoskeletal Questionnaire was used to obtain data. Data collection was carried out from March to April 2020. Data were analyzed using the SPSS version 22. Results: Cumulative prevalence of LBP was 82.9%, annual prevalence was 85.5%, while one-week prevalence of LBP was 53.6%. The factor significantly associated with LBP over the past 12 months was manual lifting of patients (*p* = 0.030). Nurses working in surgical wards had higher prevalence of LBP. About 24.7% of them changed their working unit, hospitalization was necessary for 11.9%, and 39.8% sought medical care. Conclusions: The findings from this study may better enable policymakers to adopt certain strategies toward reducing the burdens and challenges of LBP among nurses.

## 1. Introduction

Pain is an unpleasant emotional condition felt in the mind but described as occurring in a part of the body. Pain is a defense mechanism designed to make the subject protect an injured body part from additional damage. Low back pain (LBP), perhaps more accurately called lumbago or lumbosacral pain, arises below the twelfth rib and above the gluteal folds [1].

LBP is one of the most prevalent complaints necessitating health care. It is the most common type of musculoskeletal disorder (MSD) [2]. Globally, the prevalence of LBP among the general population varies from 15% to 45% [3]. In Saudi Arabia (SA), LBP prevalence is stated to be 18.8% among the general population [4].

In health-care settings, work-related illnesses and injuries have a higher prevalence compared to the general population. The reported prevalence of LBP among nurses is 85.7% in England [5], 62% in Italy [6], and 80.9% in Hong Kong [7]. In Africa, a study recorded 63.6% prevalence of LBP among nurses [8]. In 2015, a study conducted in neighboring Qatar, showed a prevalence of LBP among nurses of 54.3% [9].

Within SA, as in many other countries, LBP presents a significant issue among healthcare staff, most notably among nurses. The prevalence and risk factors of LBP in SA have been previously investigated and was comparable to levels recorded in the literature, finding that they ranged from 61.7% in Jeddah city [1] to 80% in Riyadh city [10].

Nurses most commonly experience MSDs with LBP, and they are at a higher risk of MSDs relative to other health-care workers and the general population; this is being termed as “an epidemic in nursing” [11].

The experience of LBP in relation to the inherent nature of the nursing job is typically determined by a variety of influencing factors [8]. According to recent studies, for example, LBP is often affected by sociodemographic characteristics, such as sex, age, body mass index (BMI), and experience [1,8,11,12,13,14,15,16,17,18,19,20,21]. Studies also show that conditions at the workplace, such as overtime duties, prolong working hours, working posture, and work shifts are significant predictors of LBP [9,10,14,15,16,19,22,23]. Additionally, lifestyle factors, such as obesity, and lack of physical activity, and psychological issues, such as stress and job satisfaction, have a substantial effect on incidences of LBP [8,12,18,20].

Low back pain among working nurses may influence efficiency in the clinical field. Because nurses play an important role in the health care system and represent about one-third of the workforce at any hospital, it is suspected that LBP has a direct effect on their job restrictions and attendance [23]. Hence, the present study was designed to determine the prevalence of LBP and the associated risk factors among nurses at KAUH. Understanding the prevalence of LBP and its associated risk factors in a developing country like SA, which lacks studies on this topic, is crucial. Information about this will help nursing and hospital administrators prepare effective strategies to reduce occurrences of LBP.

## 2. Materials and Methods

### 2.1. Study Design and Respondents Selection

A cross-sectional study design was performed at KAUH in Jeddah city, SA in nine major wards, medical, surgical, pediatrics, obstetrics and gynecology, operation and delivery, daycare, intensive care, and coronary care units. The study was conducted from 1 March 2020 to 30 April 2020. Both genders nurses, who provides direct patients care and were willing to participate in the study were include. Those who were pregnant, nurses with diagnosed prolapsed inter-vertebral discs, nursing interns, and students were excluded from the study.

King Abdulaziz University Hospital is a general teaching hospital in the western region of SA was inaugurated in late 1976 with a total capacity of 200 beds. Currently, the hospital’s capacity is 1067 beds. It is the largest tertiary care hospital in the western region of SA. It is well equipped with the most modern technology and staffed with about 4000 health-care professionals and administrators. This hospital serves citizens and residents alike.

The overall number of nurses in the selected area was 593. Raosoft, Inc.’s software (2004) was used for sample size calculation with response distribution at 50%, margin of error at 5%, and confidence level at 95%. Therefore, a sample size of 234 nurses was selected for this study. A convenience sample of nurses at KAUH was recruited based on the inclusion criteria to complete a written questionnaire.

### 2.2. Research Tools

The questionnaires were divided into two parts. The first part was developed by the researcher and aimed to gather information about the study participant’s background. It consisted of sociodemographic data, medical factors, and work-related factors that may contribute to the development of LBP. Sociodemographic characteristics included age, gender, nationality, marital status, and education level. Medical factors consisted of comorbid illnesses, surgical history, participant weight, and BMI. Work-related factors included current unit, working hours per day, and job experience (years of work). Other risk factors of LBP related to the work process were also investigated, such as the presence of insufficient nursing staff and manual lifting of patients. BMI is calculated by dividing the weight of the body in kilograms by the square of the height measured in meters. It is further divided into underweight [˂18.5 kg/m^2^], normal weight [18.5–24.9 kg/m^2^], overweight [25.0–29.9 kg/m^2^], and obese [≥30.0 kg/m^2^) based on to the World Health Organization’s (WHO) standards [18].

The second part was determining LBP Prevalence, the prevalence of LBP was determined using the Standardized Nordic Musculoskeletal Questionnaire for the analysis of musculoskeletal symptoms (NMQ) [24]. The questionnaire consists of structured, forced, binary, or multiple-choice variants and can be used as a self-administered questionnaire. It consists of eight questions on LBP with a picture of the body showing the possible position of LBP. The questionnaire explains cumulative life prevalence, yearly prevalence, and one-week prevalence of LBP. Moreover, the questionnaire demonstrates the consequences of LBP, such as hospitalization, change of jobs or duties, visits to a doctor or physiotherapist, and reduced activity during the last 12 months. The NMQ has been shown to be a valid and reliable instrument [24].

### 2.3. Data Collection and Statistical Analyses

The principal researcher met all the head nurses at the selected units before approaching the study participants and explaining the aim of the study. The study questionnaires were administered by the author to the nurses after providing their written consent to participate. The questionnaire took about 6 min to complete. The best time to gather data was in the afternoon and evening.

The nurses in KAUH are subjected to work either in rotating shifts or during the day (nonshift). Those who work in a rotating shift schedule are considered shift workers. The work starts at 7:00 a.m. for the morning shift and at 7:00 p.m. for the night shift. Hence, the maximum number of hours that nurses work per shift is 12 h. In contrast, those who practice day work (from 7:00 a.m. to 17:00 p.m.) are considered nonshift workers.

In the current study, the respondents with LBP is defined as ache, pain, or discomfort in the shaded area (Figure 1) whether or not it extends from there to one or both legs (sciatica) [24].

Data entry and analyses were performed using Statistical Package for Social Sciences (SPSS, version 22; IBM Corp., Armonk, NY, USA). Data were checked and explored clearly. The researchers used inferential statistical methods for the analysis. Descriptive analysis, frequency and percentage have been conducted to describe socio-demographic data, medical, work-related factors, and LBP prevalence, and correlation test by using Contingency Coefficient. Also, *p*-values ≤ 0.05 were considered significant.

### 2.4. Ethical Consideration

In the present study ethical approval was obtained from the ethical committee in the Faculty of Nursing at King Abdulaziz University (KAU) and at the King Abdulaziz University Biomedical Ethics Research Committee at KAUH (Registration No 70-20). A researcher explained the purpose of the study to the participants. Written informed consent was obtained from them. Nurses were informed that participation in this study was voluntary, and they could withdraw at any time. Moreover, confidentiality was ensured throughout the study; only the researchers had access to the data throughout the study.

## 3. Results

A total of 234 nurses fully answered the questionnaires. The response rate was 100%, and the researcher achieved good cooperation and collaboration from the hospital’s top management and participants. Table 1 presents the frequency distribution of sociodemographic characteristics of the nurses working in KAUH, who participated in the study. More than half of the participants 123 (52.6%) were between 30 and 40 years of age. The majority of nurses were female 203 (86.8%), and 178 (76.1%) were married. A high proportion of the respondents 209 (89.3%) were non-Saudi. Half of the nurses had a bachelor’s degree in nursing 117 (50.0).

Table 2 reveals the medical factors of the respondents. Most of the nurses did not complain of a medical disease history 200 (85.5%) or surgical history 158(67.5%). The majority of the participants 149 (63.7%) were classified as normal weight.

Table 3 shows the work-related factors that may influence LBP. Regarding departments, nurses working in the medical ward made up the majority group 44 (18.8%). A total of 200 (78.2%) nurses worked for more than 10 h per day. More than half of the nurses (53.8%) had more than 10 years of experience. A majority of the nurses 210 (89.7%) did manual lifting in their daily jobs. Around two-thirds 149 (63.7%) of the nurses agreed that there was not enough staff to go around.

Table 4 shows the prevalence of LBP and its characteristics among nurses. Many nurses complained of LBP, with cumulative life prevalence of LBP at 194 (82.9%), prevalence of LBP during the last 7 days at 89 (53.6%), and prevalence of LBP during the previous 12 months at 166 (85.5%). Only 28 (14.4%) of nurses didn’t had LBP during the previous 12 months. Half of the participants 97 (50.0%) had LBP that lasted from 1 to 7 days, and 15 (7.7%) of nurses reported LBP every day in the previous year.

Regarding the consequences of LBP among nurses over the past 12 months, many professional and medical consequences were reported. Most of the nurses reported that it had an effect on their professional performance, such as changing their working area 48(24.7%) and being forced to reduce their work activity 117(69.9%). Besides, LBP forced nurses to reduce their leisure activity because of LBP 76(45.8%). Low back pain prevents nurses from doing normal work at home or away from home, about 97(58.4%) of nurses experienced LPB that lasted from 1–7 days. LBP also had medical consequences, with more than one-third of nurses seeking medical advice 66(39.8%) and hospitalization being necessary for 23 (11.9%).

Table 5 displays the relationships between LBP and sociodemographic characteristics, medical factors, and work-related factors over the past 12 months. Regarding sociodemographic characteristics, there were no significant relationships between age, gender, nationality, marital status, and education level regarding LBP prevalence, with *p* > 0.05.

None of the medical factors had a significant relation with LBP prevalence among nurses, such as comorbid illnesses like diabetes or hypertension, surgical history, and weight.

There was a positive statistically significant relationship between manual lifting of patients and LBP during the last 12 months, whereas (*p* = 0.030 < α = 0.05). When being manual lifting of patients, the chance of having LBP is 87.2 % for the nurse compared to 66.7 % for non-manual lifting of patients. None of the remaining work-related factors had a significant relation with LBP prevalence among nurses, such as department, working hours, job experience, and not enough staff.

## 4. Discussion

The highest prevalence rates of work-related lower back problems were demonstrated among nurses, and LBP ranked third among musculoskeletal occupational health problems among nurses. This concern was linked to nurses’ physical activity in the hospitals and to ergonomics risk factors [1]. The present study showed a high cumulative prevalence of LBP among nurses (82.9%). This result is comparable with a previous study conducted in Jordan 78.9% [20]. The annual prevalence in this study was (85.5%); this result is in line with that of a study conducted in India 84% [16], in Riyadh 80% [10], in Jordan 78.9% [20], in Malaysia 74.8% [22], in Bahrain 73.5% [17], and in Bangladesh 72.9% [19]. However, the one-year prevalence was lower in Western Ethiopia 63.6% [8], in Sarawak 63.1% [21], and in Tunisia 58.1% [12]. The one-week prevalence was 53.6%, similar to Ethiopia 53.4% [8]. Only the one-week prevalence in Slovenia was found to be lower 37.6% [11]. The high prevalence of LBP amongst nurses emphasizes the importance of the problem of back pain in nursing personnel. The differences in LBP prevalence rates reported in the literature across countries can be attributed to the nature of the workplace and variations in the methodological approach used for the measurement of LBP prevalence. Cultural differences might also influence respondents’ willingness to report LBP and tolerance of pain [21].

The current study showed that the majority of participants were married 178 (76.1%). Additionally, the results revealed a high prevalence of LBP among married participants 121 (72.9%) as compared to single and divorced nurses and widows. Because of cultural beliefs, women, especially married women, are exposed to strenuous activities and household activities such as daily and nightly routine domestic tasks that involve taking care of their families besides doing their job-related activities. These consequently increase their risk of suffering LBP. This is comparable with a study that reported that 69.1% of married women complained of LBP [15]. There was no significant relation between prevalence of LBP and marital status in this study, which is in agreement with the literature [9,12,15,18].

The type of ward that nurses work in can contribute to high LBP rates. This study found that nurses working in the surgical ward had the highest LBP prevalence as compared to nurses in the medical ward and other wards—although the difference was not significant. This is possibly linked with the higher physical workload and the amount of work pressure preoperative and postoperative patients create. They require more assistance with moving in bed and with transfers in the surgical department. The findings of the current study correspond with the results of previous study, indicating that nurses working in surgical-discipline wards have a higher prevalence of LBP than those working in medical-discipline wards [21]. As a result, it was suggested that nurses must be rotated in their workplace to provide a balance level [14].

Total years of nursing experience or length of employment is considered an important predictor of LBP among nurse professionals. In the current study, it was observed that LBP prevalence was higher among nurses who had been in the profession for more than ten years 96 (88.9%). This can be explained by the fact that nurses are exposed to more events involving inappropriate use of the back mechanism and accumulated back stress with more years of practice [20]. This finding was consistent that of several studies that showed a significant correlation between the prevalence of LBP and job experience and revealed that employees with a comparatively longer period of employment were more likely to be susceptible to back pain complaints compared to workers with a shorter period of employment [11,13,15,18,20,21]. In contrast, another study observed that LBP was more prevalent among nurses with less than five years of nursing experience than in those with five years or more of nursing experience. Workers with a comparatively lower period of service may typically lack awareness and skills about safety procedures and hazard control mechanisms, making them prone to injuries and accidents at the workplace [8]. However, the total number of working years did not have a significant effect on LBP among nurses in the recent study.

Low back pain was reported to increase in parallel with the increase in working hours, and this result was linked with the decrease in rest time [1]. The current study shows that majority of the nurses who worked more than ten hours per shift experienced LBP 127 (86.4%)—although the difference was not significant. This finding is in agreement with the results of previous studies, which showed that nurses with twelve hour shifts had a significantly higher prevalence rate of LBP than those with a shift time of just eight hours [9]. Additionally, a study found that nurses who work more than seven hours per day have been significantly associated with LBP [22].

This study found that there was a positive statistically significant relationship between manual lifting of patients and LBP (*p* = 0.030). When being manual lifting of patients, the chance of having LBP is 87.2% for the nurse compared to 66.7% for non-manual lifting of patients. With long working hours and high workload, nurses expose themselves to repeat manual handling and, therefore, sustained accumulation of wear and tear of the back muscle and putting them at risk for LBP development. In addition, some patients may be combative or uncooperative. Any unpredictable movement or resistance from the patient may throw the nursing personnel off balance during the transfer, resulting in a back injury [15]. This result is consistent with one study found that nurses who do manual lifting in their daily jobs consider it an occupational factor associated with LBP [19]. The finding was also confirmed by a study showed that nurses with LBP had a higher frequency in manual handling of patients [21]. Further, manual patient handling tasks in wards was found to be a significant risk factor for LBP occurrences [22].

In the recent study, more than one-third of nurses 66 (39.8%) sought medical advice for their back pain, whether by visiting a doctor or a physiotherapist, and 23 (11.9%) of participants had been admitted to hospital for their back pain. Similarly, a study found that 44% of nurses sought medical advice, and hospitalization was necessary for 8.6% of nurses [20]. This may be because they are in a medical setting, so they can easily get examinations and ask for prescriptions from their physician colleagues.

This study has some limitations that should be acknowledged. It was conducted among staff nurses in one hospital; therefore, extensive generalization cannot be made. Additionally, psychological factors and their relation with LBP were not investigated in this study—although the study successfully provided significant information about the prevalence of LBP and risk factors to which nurses are exposed, which will help in addressing this issue and provide a better working atmosphere for nurses.

It is recommended for the hospital administrations to design effective interventions and adopt certain strategies to improve the condition of LBP and its ensuing effects among nurses in KAUH such as, regular in-service training on back care and ergonomics must be conducted in various wards to assist nurses in refreshing their manual handling technique knowledge. Nurses should take responsibility for their health by taking into account preventive measures and coping strategies against work-related injuries, such as LBP. For example, nurses must ask for assistance when performing patient handling activities and should perform relaxing and stretching exercises during work hours. Further studies are required to evaluate the prevalence of LBP and more comprehensive risk factors are needed to identify ways of providing a healthy and safe working environment for nurses.

## 5. Conclusions

Low back pain is a serious health problem affecting nurses and they should give importance to their own well-being. This will, in turn, ensure the best quality of care is delivered to patients. In the present study, LBP was highly prevalent among nurses in KAUH, which could be attributed to the nature of their work. Manual lifting of patients was the only risk factor statistically significantly associated with LBP among nurses at KAUH. The findings of the current study alarming and point to a need for solutions and certain strategies should be adopted toward reducing the burdens and challenges of LBP such as, nurses must ask for assistance when performing patient handling activities, scheduling adequate rest breaks and doing relaxing and stretching exercises during work hours. The results are hoped that this study will provide the groundwork for more elaborated and elucidative studies in the future.

## Figures and Tables

**Figure 1 ijerph-18-01567-f001:**
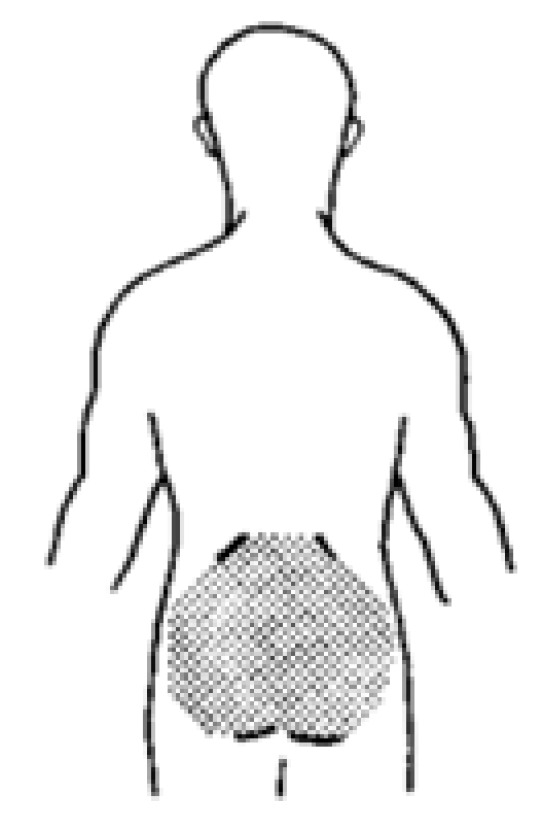
Region of low back pain. Adapted from (Kuorinka et al., 1987).

**Table 1 ijerph-18-01567-t001:** Sociodemographic Characteristics of the Respondents.

Variables	Respondents (*n* = 234)
	*n* (%)
**Age**	
<30	45 (19.2)
30–40	123 (52.6)
>40	66 (28.2)
**Gender**	
Male	31 (13.2)
Female	203 (86.8)
**Nationality**	
Saudi	25 (10.7)
Non-Saudi	209 (89.3)
**Marital status**	
Single	54 (23.1)
Married	178 (76.1)
Divorced	1 (0.4)
Widow	1 (0.4)
**Education Level**	
Diploma	113 (48.3)
Bachelor	117 (50.0)
Post-graduate	4 (1.7)

**Table 2 ijerph-18-01567-t002:** Medical Factors of the respondents.

Variables	Respondents (*n* = 234)
	*n* (%)
**Co-morbid illness**	
Yes	34 (14.5)
No	200 (85.5)
**Surgical history**	
Yes	76 (32.5)
No	158 (67.5)
**Weight**	
Underweight	9 (3.8)
Normal weight	149 (63.7)
Overweight	69 (29.5)
Obese	7 (3.0)

**Table 3 ijerph-18-01567-t003:** Work-related Factors of the respondents.

Variables	Respondents (*n* = 234)
	*n* (%)
**Departments**	
Medical ward	44 (18.8)
Surgical ward	41 (17.5)
Operation rooms	36 (15.4)
Obstetrics & gyn.	9 (3.8)
Delivery rooms	14 (6.0)
Intensive care unit	27 (11.5)
Coronary care unit	15 (6.4)
Pediatric ward	35 (15.0)
Day care unit	13 (5.6)
**The Working hours per shift**	
>10 h	51 (21.8)
8–10 h	183 (78.2)
**Job experience (years of work)**	
<1 y	4 (1.7)
1–5 y	42 (18.0)
6–10 y	62 (26.5)
	126 (53.8)
**Manual lifting**	
Yes	210 (89.7)
No	24 (10.3)
**Not enough staff**	
Yes	149 (63.7)
No	85 (36.3)

**Table 4 ijerph-18-01567-t004:** Prevalence of LBP and its consequences among nurses.

Variables	Respondents (*n* = 234)
	*n* (%)
**Cumulative life prevalence**	
Yes	194 (82.9)
No	40 (17.1)
**Annual prevalence**	
0 day	28 (14.4)
1–7 days	97 (50.0)
8–30 days	16 (8.2)
>30 days	38 (19.6)
Every day	15 (7.7)
**One week prevalence**	
Yes	89 (53.6)
No	77 (46.4)
**Change jobs or duties**	
Yes	48 (24.7)
No	146 (75.3)
**Reduce work activity**	
Yes	117 (69.9)
No	50 (30.1)
**Reduce leisure activity**	
Yes	76 (45.8)
No	90 (54.2)
**Prevented to perform normal work**	
0 day	35 (21.1)
1–7 days	97 (58.4)
8–30 days	16 (9.6)
>30 days	18 (10.8)
**Visiting doctor, physiotherapy or chiropractor**	
Yes	66 (39.8)
No	100 (60.2)
**Admitted to hospital**	
Yes	23 (11.9)
No	171 (88.1)

**Table 5 ijerph-18-01567-t005:** Relationships between low back pain and risk factors among nurses in KAUH over the past 12 months.

Variables	Respondents with LBP		Respondents without LBP		Contingency Coefficient	*p*-Value
	N	Row-%	N	Row-%		
**Age**						
<30	29	85.3	5	14.7	0.005	0.998
30–40	90	85.7	15	14.3		
>40	47	85.5	8	14.5		
**Gender**						
Male	22	88.0	3	12.0	0.027	0.711
Female	144	85.2	25	14.8		
**Nationality**						
Saudi	20	95.2	1	4.8	0.095	0.182
Non-Saudi	146	84.4	27	15.6		
**Marital status**						
Single	43	91.5	4	8.5	0.106	0.532
Married	121	83.4	24	16.6		
Divorced	1	100.0	0	0.00		
Widow	1	100.0	0	0.00		
**Education Level**						
Diploma	79	84.0	15	16.0	0.068	0.633
Bachelor	83	86.5	13	13.5		
Post-graduate	4	100.0	0	0.00		
**Co-morbid illness**						
Yes	26	92.9	2	7.1	0.085	0.235
No	140	84.3	26	15.7		
**Surgical history**						
Yes	55	83.3	11	16.7	0.046	0.525
No	111	86.7	17	13.3		
**Weight**						
Underweight	7	100.0	0	0.0	0.138	0.285
Normal weight	98	82.4	21	17.6		
Overweight	54	88.5	7	11.5		
Obese	7	100.0	0	0.0		
**Departments**						
Medical ward	33	80.5	8	19.5	0.184	0.558
Surgical ward	34	94.4	2	5.6		
Operation rooms	30	90.9	3	9.1		
Obstetrics & gyn.	5	71.4	2	28.6		
Delivery rooms	11	91.7	1	8.3		
Intensive care unit	19	82.6	4	17.4		
Coronary care unit	7	87.5	1	12.5		
Pediatric ward	21	80.8	5	19.2		
Day care unit	6	75.0	2	25.0		
**The Working hours per shift**						
>10 h	127	86.4	20	13.6	0.042	0.562
8–10 h	39	83.0	8	17.0		
**Job experience (years of work)**						
<1 y	2	100.0	0	0.0	0.124	0.386
1–5 y	30	83.3	6	16.7		
6–10 y	38	79.2	10	20.8		
>10 y	96	88.9	12	11.1		
**Manual lifting**						
Yes	156	87.2	23	12.8	0.154	0.030
No	10	66.7	5	33.3		
**Not enough staff**						
Yes	114	89.1	14	10.9	0.137	0.054
No	52	78.8	14	21.2		

## Data Availability

The data is available from the authors on request.

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
