# Peer review of "Prevalence of Low Back Pain and Associated Risk Factors among Nurses at King Abdulaziz University Hospital"

_ijerph, 2021, doi:10.3390/ijerph18041567_

Round 1

Reviewer 1 Report

This research article finished by ALMaghrabi and ALSharif, analyze the prevalence of low back pain (LBP) among nurses at the King Abdulaziz university hospital through demographic data and questionnaires. The author did careful sampling and detailed analysis, it reminds the occupation of nurses that there is a risk of LBP and provides help to improve the quality of nurses’ work.

There are several issues:

1 The author analyzes the nationality factor in table 1 and 5, how about the Race, like Arab, Indian, east Asian. This comparison is more meaningful than nationality.

2 In table 3 Work related factors, are the research subject engaged in other part time job or intense sport exercise?

Author Response

Thanks for your comments. For point regarding the race and nationality comparison this will be considered in the next future studies. However, for the aim of this study race is not main factor to be study in depth.

For second point, all subjects in study were recruited for full time job in the hospital per the signed contract. For intense sport exercise was not considered to be measured in this study as not a major factor that may affect the study.  Moreover, intense sport exercise in the literature was not a significant factor that may cause back pain among nurses or health care provider. 

Reviewer 2 Report

The paper assesses the prevalence of the low back pain among nurses, whose professional tasks could favour the spinal problems. The paper can be of interest to the journal’s readers, but some issues should be addressed prior to its publication.

  1. The number of wards from which the nurses were recruited. The authors gave no 8 in the Abstract, and 9 in the Methods.
  2.  In the Methods section the authors stated that “Data were checked, explored, and cleaned”. The use of word “cleaned” is rather disturbing, as it can suggest that the data which were outliers were removed from the database. Some outliers are data which should be removed (errors, coming from subjects who should be excluded from the study, etc.) but some outliers reflect the examined phenomena and should be kept in the database. Some explanation about “cleaning” the database should be added.
  3. The interesting finding was that the low back pain was more frequent among non-Saudi nurses. In the Discussion section the authors suggest, that this difference may be attributed to the genetic factors. It would be interesting to explore other factors: prolonged stress connected with living in foreign country with different culture, and far from the family for example. Prolonged stress can result in increased muscle tone of the muscles, and thus can lead to the back pain.
  4. Total years of nursing was found to be an important factor of low back pain, in contrast to age. This is for me a little surprising, as I would assume that older nurses worked longer in the profession. Some explanation should be added to the Discussion section.

Author Response

Comment # 1 : The number of wards were corrected in the abstract.

Comment # 2 : The "cleaning" was removed to make the article clearer to the readers and not confused.

Comment # 3:The other factor such as prolonged stress connected with living in foreign country with different culture, and far from the family would be consider in the next future study. Really interesting and thanks

Comment # 4: In the discussion paragraph # 6 the comment was explained  briefly. It was really very surprising to us as researchers to get this significant results.  We were planning investigate more regarding to this point or results. 

Reviewer 3 Report

The document focuses on a particular population and a specific job, but I think to understand that this is a category that needs more attention than now, so I hope that this is the first step in a change's way.

The paper is well written and fleunt in reading.

Only one observation about the figures, if the values between round brackets are standard deviations it coud be better to write these informaton in the captions and add in the brackets this symobl ±.

Author Response

Comment # 1 : brackets were added. 

Reviewer 4 Report

The manuscript describes what factor is important to generate low back pain and relative environmental condition in hospitals worker. In the study of the correlation, the authors suggest that the genetic difference and working hours per shift is a key point for the generation of LBP. It is slightly different from previous studies which are suggested the working load is a critical factor to develop the LBP. However, the manuscript needs to fix the table format and logical concerns.

  1. In Table 5, the ordering of the rows is inconsistent, making it less readable. It is better to fix the table format.
  2. LBP is more common among non-Saudi nurses 146(88.0%). This could be a result of genetic traits ~ LBP symptoms. This sentence is not logically acceptable because the manuscript did not show enough possible evidence for supporting results or possible cleared references such as SNP or genetic difference. I think, it is a more logically valid approach to check whether there is a relationship between races (Saudi vs non-Saudi) for supporting the main idea with factors such as "The Working hours per shift" which are statistically significant differences in Table 5.

Author Response

1- Comment #1 : table was edited 

2-Comment #2 : I agree, genetic trait was removed and replaced by working hours per shift.